# Healthcare resource utilisation and cost of pneumococcal disease from 2003 to 2019 in children ≤17 years in England

**Salini Mohanty**[1]*, **Bélène Podmore**[2], **Ana Cuñado Moral**[2], **Ian Matthews**[3], **Eric Sarpong**[4], **Agueda Azpeitia**[2], **Nawab Qizilbash**[2]

**1** Merck & Co., Inc, Center for Observational and Real-World Evidence (CORE), Rahway, New Jersey, United States of America, **2** OXON Epidemiology Ltd, Epidemiology & Statistics, Madrid, Spain, **3** MSD (UK) Ltd, Value, Access and Devolved Nations (VAD), London, United Kingdom, **4** Merck & Co., Inc., Real-world Data Analytics and Innovation (RDAI), Rahway, New Jersey, United States of America

* salini.mohanty@merck.com

## Abstract

### Objective

To estimate healthcare resource utilisation (HCRU) and costs associated with pneumococcal disease (PD) in children aged ≤17 years in England from 2003–2019.

### Methods

A retrospective study in children aged ≤17 years was conducted using the Clinical Practice Research Datalink Gold primary care database and Hospital Episodes Statistics Admitted Patient Care database from 2003–2019. Episodes of invasive pneumococcal disease (IPD) were identified in hospital, pneumococcal pneumonia (PP) and all-cause pneumonia (ACP) episodes in primary care and in hospital, and acute otitis media (AOM) episodes in primary care. General practitioner (GP) visits and inpatient admission yearly rates were calculated per 1,000 persons. The average inpatient and primary care cost per episode were calculated. The Mann-Kendall test was used to assess monotonic time trends.

### Results

1,500,686 children were followed from 2003–2019. The highest average inpatient cost per episode [£34,255 (95%CI 27,222–41,288)] was in IPD, followed by ACP [£3,549 (95%CI 3,405–3,693)] and PP [£1,498 (95%CI 1,153–1,843)]. The highest primary care costs per episode were in AOM [£48.7 (95%CI 48.7–48.7)], followed by PP [£38.4 (95%CI 37.0–39.7)] and ACP [£28.6 (95%CI 28.2–29.1)]. The highest inpatient admission and GP visits yearly rates were observed in children aged <2 years. Across years, a significant decrease in GP visits yearly rates was observed for PP, ACP and AOM in children overall (p-value<0.001). A decrease in primary care costs was observed for ACP (p-value<0.001). There was an increasing trend in AOM primary care costs (p-value<0.001). No significant trends were observed in inpatient admission yearly rates in PP, ACP or IPD and inpatient costs per episode in PP, ACP and IPD.

Episode Statistics Admitted Patient Care (HES APC), a real-world research service providing anonymised linked primary care and secondary care data. The licencing agreement between Merck Sharp & Dohme LLC, a subsidiary of Merck & Co., Inc., Rahway, NJ, USA and CPRD, and the data governance of CPRD prevent the distribution or availability of sensitive patient data to other persons. The datasets generated during and/or analysed during the current study are not publicly available. Access to the data are available from CPRD for researchers who meet the criteria for access via CPRD's Research Data Governance (RDG) Process (details at https://cprd.com/dataaccess).

**Funding:** This research was sponsored and funded by Merck Sharp & Dohme LLC, a subsidiary of Merck & Co., Inc., Rahway, NJ, USA. Salini Mohanty, Ian Matthews and Eric Sarpong are employees of Merck Sharp & Dohme LLC, a subsidiary of Merck & Co., Inc., Rahway, NJ, USA. The funders contributed in study design, decision to publish, and preparation of the manuscript.

**Competing interests:** I have read the journal's policy and the authors of this manuscript have the following competing interests: Salini Mohanty, Ian Matthews, and Eric Sarpong are employees of Merck Sharp & Dohme LLC, a subsidiary of Merck & Co., Inc., Rahway, NJ, USA and may own stock/stock options in Merck & Co., Inc., Rahway, NJ, USA. Bélène Podmore, Ana Cuñado, Agueda Azpeitia, and Nawab Qizilbash are employees of OXON Epidemiology Ltd, Epidemiology & Statistics, Madrid, Spain, an independent contract research organization, that received funding from Merck Sharp & Dohme LLC, a subsidiary of Merck & Co., Inc., Rahway, NJ, USA to design and conduct this study.

**Abbreviations:** ACP, All-cause pneumonia; AOM, Acute otitis media; CI, Confidence interval; CPRD, Clinical Practice Research Datalink; GP, General practitioner; HCRU, Healthcare resource utilisation; HES-APC, Hospital Episode Statistics Admitted Patient Care; HIV, Human Immunodeficiency Virus; HRG, Healthcare resource groups; ICD-10, International Statistical Classification of Diseases and Related Health Problems 10th revision; IMD, Index of multiple deprivation; IPD, Invasive pneumococcal disease; IQR, Interquartile range; ISAC, Independent Scientific Advisory Committee; JCVI, Joint Committee on Vaccination and Immunisation; Max., Maximum; Min., Minimum; NHS, National Health Service; PCV, Pneumococcal conjugate vaccine; PCV13, 13-valent Pneumococcal Conjugate Vaccine; PCV7, 7-valent Pneumococcal Conjugate Vaccine; PD,

## Conclusion

From 2003–2019, primary care HCRU and costs decreased (except for PP cost), but no trends in inpatient HCRU and costs were observed. The economic burden of pneumonia, IPD and AOM remains substantial in children aged ≤17 years in England.

## Introduction

Pneumococcal disease (PD), caused by the bacterium *Streptococcus pneumoniae (S. pneumoniae)*, can present as non-invasive or invasive infections. Non-invasive disease includes middle ear infections (otitis media), sinusitis and mild respiratory tract mucosal infections, while invasive pneumococcal disease (IPD) occurs when the bacterium enters a normally sterile site and includes septicaemia, bacteraemic pneumonia and meningitis [1–3]. PD is a leading cause of bacterial pneumonia, meningitis and sepsis worldwide [4] and causes significant morbidity, mortality, and economic burden [5].

It is estimated that there are 709 million acute otitis media (AOM) cases each year globally, with 51% of these occurring in children under five years old [6]. AOM is most commonly seen between the ages of 6 to 24 months [7], and is a leading indication for antibiotic treatment in Europe and the US [1, 8]. In a systematic review assessing the predominant bacterial pathogens that cause AOM in children found the percentage of bacteria detected from middle ear fluid in Europe were as follows: 30.2% [18.8 (min), 49.4 (max)] *S. pneumoniae*; 23.3% [11.8 (min), 29.1 (max)] *Haemophilus influenzae* and 11.6% [1.0 (min), 27.8 (max)] *Moraxella catarrhalis* [9]. Between 2 million and 2.6 million cases of pneumonia are estimated to occur yearly in children aged <5 years who reside in developed countries [10]. According to the World Health Organization (WHO), *S. pneumoniae* is the most common cause of bacterial pneumonia in children [4]. Estimates of the percentage of these children who are likely to be hospitalised are highly variable and likely depend on local factors such as clinical practices, cultural attitudes and health insurance initiatives. In developed countries, the number of children hospitalised with pneumonia ranges from 100,000 to 720,000 children per year [10].

Young children, the elderly and people in clinical risk groups are most at risk of severe PD, and are therefore recommended pneumococcal immunisation [3]. In the UK, the 7-valent pneumococcal conjugate vaccine (PCV7) was introduced into the routine childhood immunisation program in 2006, for children aged 2, 4, and 12–13 months, alongside a 12-month catch-up program for children younger than 2 years. In April 2010, the 13-valent PCV (PCV13) replaced PCV7, administered at 2 months and 4 months with a booster at 12–13 months of age [11, 12]. In 2019, the UK Joint Committee on Vaccination and Immunisation (JCVI) recommended to switch from a 2+1 to 1+1 schedule. Thus, all healthy infants born on or after 1 January 2020 receive PCV13 at 3 months followed by a booster at 12–13 months [12, 13]. Potential advantages of reducing the number of doses of PCV from three to two in the infant immunisation schedule may include making space in the infant programme for additional vaccines in development and cost reduction [14]. In 2019/20, 93.2% of children were vaccinated by their first or second birthday in England [15].

Worldwide, there has been a decrease in the overall incidence of PD, following the introduction of PCV7 and PCV13 [11, 16–21]. In countries with high vaccination coverage and a mature infant vaccination programme, the epidemiology of *S. pneumoniae* now largely reflects infections caused by a limited number of serotypes that are not covered by the current infant pneumococcal vaccines [22]. Nevertheless, there is still some persistence in PCV13 serotypes,

Pneumococcal disease; PP, Pneumococcal pneumonia; PY, Person-years; S. pneumoniae, *Streptococcus pneumoniae*; UTS, Up-to-standard; WHO, World Health Organization.

in particular serotypes 3 and 19A [23]. Serotype replacement is also highly variable across countries [17].

There is limited information on the impact of PCV vaccination on healthcare resource utilisation (HCRU) and costs due to PD in children in the UK. The current published literature provides information mainly in adults and the elderly, while specific studies focused on children are lacking. Next-generation PCVs are now in late-stage clinical development for paediatric use, including a 15-valent PCV recently approved for use in adults and children in Europe, the US and Canada [24–27], and a 20-valent PCV approved for use in adults in Europe [28] and the US [29]. It is therefore important to understand and quantify the economic burden of PD in children before the introduction of new higher-valent PCVs in England. Thus, the aim of this study is to describe the HCRU and costs of PD in England, in children aged 0–17 years across different PCV periods, from 1 January 2003 to 31 December 2019.

## Materials and methods

### Study design

This retrospective observational cohort study was conducted using linked data from the Clinical Practice Research Datalink (CPRD)-Gold and the Hospital Episodes Statistics Admitted Patient Care (HES-APC) databases [30, 31]. The study population included children aged ≤17 years in England from 1 January 2003 to 31 December 2019. For children aged 1–17 years, the eligible criteria for inclusion were: (1) at least 12 months of medical up-to-standard practice [UTS; measure of data quality as defined in CPRD [31]], (2) last practice data collection date and (3) study inclusion date to ensure their medical history could be assessed with continuous follow-up in the preceding last 6 months. Continuous follow-up was defined from the current registration date and only for patients with no follow-up interruptions or single interruptions ≤7 days. Children aged <1 year did not need to meet these criteria. The earliest of these events defined the start of the follow-up period: (a) the start of study period (1 January 2003), (b) the date of birth, or (c) the start of data collection. The end of study period was defined by the earliest of these events: (a) the end of the study period (31 December 2019), (b) the end of the year in which the patient turns 17 years, (c) death, (d) transfer out of the practice or (e) end of data collection.

### Data source

Children were identified using linked primary and secondary healthcare data (from the CPRD Gold and the HES APC databases). The CPRD-Gold database consists of anonymised medical records from general practitioner (GP) practices as part of routine clinical care in England, Wales, Scotland and Northern Ireland, in the UK [31]. The CPRD Gold database currently has 3.11 million acceptable patients (i.e., registered at currently contributing practices, excluding transferred out and deceased patients) from 985 practices, representing 4.64% of the UK population [32]. Patients are representative of the general population in the UK, in terms of age, sex and ethnicity [31]. The CPRD primary care database is a well-established, reliable source of health data for research, and includes data on demographics, symptoms, diagnoses, tests, health-related behaviours, therapies and referrals to secondary care [31]. Since 1997, the HES-APC database includes all inpatient admissions to National Health Service (NHS) hospitals in England. The HES-APC information collected includes primary and secondary diagnoses, procedural events, and dates of admission and discharge from hospital [30].

## Exposures

Four diseases were defined in this study: invasive pneumococcal disease (IPD), pneumococcal pneumonia (PP), all-cause pneumonia (ACP) and acute otitis media (AOM) episodes. The diagnosis codes used are presented in S1 Table in S1 File.

IPD was defined as inpatient episodes caused by *S. pneumoniae*, and includes pneumococcal bacteraemia/septicaemia, pneumococcal meningitis, pneumococcal bacteraemic pneumonia and other IPD manifestations (e.g., pericarditis, osteomyelitis) [1, 2, 21, 33]. IPD was identified using the International Statistical Classification of Diseases and Related Health Problems 10th revision (ICD-10) diagnosis codes in the HES-APC database.

Pneumonia episodes were classified as PP and ACP. Pneumonia was captured in both primary care and in the inpatient setting from linked CPRD-Gold and HES-APC databases, using Read and ICD-10 diagnosis codes, respectively. PP referred to pneumonia caused by *S. pneumoniae*. ACP included pneumonia caused by all known and unknown pathogens (bacterial, viral, and fungal), including pneumonia caused by *S. pneumoniae*. This wider definition was included as an estimate of the burden of PD as diagnostic tests to identify causative pathogen are infrequently performed in clinical practice [9]. The diagnosis and coding of PP would therefore likely underestimate disease burden [34, 35]. This approach has been used previously [21, 36–38].

AOM episodes were identified in primary care (CPRD-Gold) using Read diagnosis codes. AOM consists of episodes caused by all known and unknown pathogens, including suppurative AOM, suppurative otitis media and unspecified otitis media.

A gap of 90 days between inpatient episodes with PP, ACP and IPD diagnosis codes defined the start of a new episode [21, 33]. For PP, ACP and AOM GP episodes in which there was only a single visit with a Read diagnosis code for pneumonia or AOM, a period of 14 days was considered after that visit to account for antibiotic treatment.

## Outcomes & covariates

HCRU and costs were evaluated for IPD, pneumonia and AOM over the study period. The HCRU measures included number of GP visits, number of GP antibiotic prescriptions, number of hospital admissions and average length of hospitalisation. Measures of cost included primary care cost (GP visit + GP antibiotic prescription), inpatient cost, and total cost (primary care + inpatient). Both costs per episode and the total annual costs were estimated.

The following population characteristics were assessed: age group, sex, geographic region, urbanicity, social deprivation, ethnicity and risk factors. The age groups were <2, 2–4 and 5–17 years old. For England, there are ten geographic regions in CPRD: North East, North West, Yorkshire & The Humber, East Midlands, West Midlands, East of England, South West, South Central, London, and South East Coast [31]. The Rural-Urban classification was updated in 2011, defining areas as rural if they fall outside of areas forming settlements with populations of at least 10,000 [39, 40]. The 2019 English Index of Multiple Deprivation score (IMD) was used to measure social deprivation. IMD describes data in quintiles, with quintile 1 denoting the least deprived areas and quintile 5 the most deprived areas [39, 41]. Risk factors were defined based on the Green Book chapter Pneumococcal Disease [42], and included asplenia or dysfunction of the spleen, chronic respiratory disease, chronic heart disease, chronic kidney disease, chronic liver disease, diabetes, immunocompromising diseases (such as Human Immunodeficiency Virus (HIV) or other immunodeficiencies), cerebrospinal fluid leak and cochlear implant.

**Table 1. Baseline characteristics of the study population (n = 1,500,686) from 2003–2019.**

| | N | %[a] |
|---|---|---|
| **Study population by year** | | |
| 2003 | 527,885 | - |
| 2004 | 579,904 | - |
| 2005 | 627,223 | - |
| 2006 | 668,922 | - |
| 2007 | 707,332 | - |
| 2008 | 745,655 | - |
| 2009 | 766,870 | - |
| 2010 | 790,398 | - |
| 2011 | 790,747 | - |
| 2012 | 784,472 | - |
| 2013 | 772,412 | - |
| 2014 | 699,879 | - |
| 2015 | 585,480 | - |
| 2016 | 430,976 | - |
| 2017 | 341,078 | - |
| 2018 | 282,628 | - |
| 2019 | 243,745 | - |
| **Age (years)** | | |
| Minimum, maximum | 0.0, 17.0 | - |
| Mean, standard deviation | 5.7, 5.7 | - |
| Median (lower quartile, upper quartile) | 4.0 (0.0–11.0) | - |
| **Age group** | | |
| <2 years | 555,367 | 37.0 |
| 2–4 years | 215,630 | 14.4 |
| 5–17 years | 729,689 | 48.6 |
| **Sex** | | |
| Male | 775,231 | 51.7 |
| Female | 725,455 | 48.3 |
| **Geographic region** | | |
| North East | 29,145 | 1.9 |
| North West | 224,670 | 15.0 |
| Yorkshire & The Humber | 52,502 | 3.5 |
| East Midlands | 45,708 | 3.0 |
| West Midlands | 173,151 | 11.5 |
| East of England | 154,621 | 10.3 |
| South West | 175,419 | 11.7 |
| South Central | 191,935 | 12.8 |
| London | 231,471 | 15.4 |
| South East Coast | 222,064 | 14.8 |
| **Urbanicity** | | |
| Urban | 1,314,679 | 87.6 |
| Rural | 186,007 | 12.4 |
| **Social deprivation (IMD Score)** | | |
| Quintile 1 (least deprived) | 322,501 | 21.5 |
| Quintile 2 | 293,616 | 19.6 |
| Quintile 3 | 297,302 | 19.8 |

(*Continued*)

**Table 1.** (Continued)

|  | **N** | **%ᵃ** |
|---|---|---|
| Quintile 4 | 289,738 | 19.3 |
| Quintile 5 (most deprived) | 295,851 | 19.7 |
| Missing | 1,678 | 0.1 |
| **Ethnicity** |  |  |
| White | 270,010 | 18.0 |
| South Asian | 43,354 | 2.9 |
| Black | 27,871 | 1.9 |
| Other | 14,403 | 1.0 |
| Mixed | 18,183 | 1.2 |
| Not stated / Missing / Inconsistent | 1,126,865 | 75.1 |
| **Risk factors** |  |  |
| No history of any risk medical condition | 1,398,469 | 93.2 |
| History of any risk medical conditionᵇ | 102,217 | 6.8 |
| Asplenia or dysfunction of the spleen | 810 | 0.1 |
| Chronic respiratory disease | 92,424 | 6.2 |
| Chronic heart disease | 3,806 | 0.3 |
| Chronic kidney disease | 585 | 0.0 |
| Chronic liver disease | 162 | 0.0 |
| Diabetes | 1,946 | 0.1 |
| Immunocompromising diseases | 10,579 | 0.7 |
| Cerebrospinal fluid leak | 38 | 0.0 |
| Cochlear implant | 134 | 0.0 |

ᵃSome totals sum to more or less than 100% due to rounding.

ᵇThe sum of each of the risk medical conditions does not match the total number of any risk medical condition because a patient could have more than one medical condition. IMD: Index of Multiple Deprivation; N: number.

## Statistical analyses

All analyses were descriptive and conducted for all children ≤17 years, and separately by age groups (<2, 2–4, and 5–17 years). HCRU was calculated as GP visits, GP antibiotic prescriptions and inpatient admission yearly rates per 1,000 persons, as the total amount of each resource used, divided by the total amount of time with the disease per year. The average length of hospitalisation was calculated as the mean days of hospitalisation per episode. Rates and days of hospitalisation are presented with two-sided 95% confidence intervals (CI) calculated assuming a Poisson distribution.

The evaluation of costs was based on a perspective from the Healthcare System and focused on direct medical costs and the average cost of GP antibiotic prescriptions. Thus, the total cost resulted from the use of healthcare resources after the imputation of monetary values to the units of resource used.

The average cost per episode and total annual cost included: PP, ACP and AOM primary care cost (GP visit + GP antibiotic prescription); IPD, PP and ACP inpatient cost, and PP and ACP total cost (primary care + inpatient). The information on prices, tariffs, and unit costs of the healthcare resources of interest as well as for the Healthcare resource groups (HRGs) for the study were retrieved and updated to the date when the data analysis started. This information was sourced from existing British National Guidelines [NICE Guidelines Otitis media (acute): antimicrobial prescribing [43] and pneumonia (community-acquired): antimicrobial

prescribing [44]] and resources for costing, publicly available repositories, published prices in previous articles with health economic evaluations, and proprietary healthcare cost databases with accredited use in publications, and pricing and reimbursement dossiers [45–48]. These reference costs are presented in S2 Table in S1 File. Expenditures were standardised to 2019 UK pounds by using the Consumer Price Index medical care component from the year when the price was documented.

To assess whether there were any significant monotonic changes in HCRU and costs for each manifestation over the study period, the Mann-Kendall linear trend test was used. All analyses were completed using SAS version 9.4 (SAS Institute, Inc., Cary, North Carolina).

### Ethics statement

All methods were performed in accordance with the relevant guidelines and regulations and the protocol (protocol ID: 20_000260) was approved by the CPRD Independent Scientific Advisory Committee (ISAC). Informed consent was not required as the CPRD database provides anonymized data from medical records.

## Results

The study population included 1,500,686 children aged ≤17 years, who experienced an episode of PD from 2003 to 2019. The study population characteristics at their inclusion in the study are shown in Table 1. 48.3% of the children were female and the median age was 4.0 years (interquartile range, IQR 0.0–11.0). The largest group of children (48.6%) were between the ages 5–17 years and lived in urban areas (87.6%). By geographical region, the most populated regions were London (15.4%), the North West (15.0%), and the South East Coast (14.8%). The study population was mostly evenly distributed with regard to social deprivation, although a slightly higher proportion resided in the quintile representing the least deprived (quintile 1). Having a history of any of the selected at-risk medical conditions was uncommon —only 6.8% of children had an at-risk medical condition. 75.1% of the children had "not stated/missing/inconsistent information" on ethnicity.

### IPD

From 2003 to 2019, 170 IPD inpatient episodes were observed, with more than half (60%) of episodes occurring in children aged <2 years. Table 2 summarises the IPD HCRU across the study period for children overall and stratified by age group. The IPD inpatient admission yearly rate in children ≤17 years was 0.04 (95% CI 0.03–0.04) per 1,000 patients. Differences in the IPD inpatient admission yearly rate were observed by age group, 0.28 (95% CI 0.24–0.33) in children aged <2 years, 0.04 (95% CI 0.03–0.05) in children aged 2–4 years, and 0.01 (95% CI 0.01–0.01) in children aged 5–17 years (per 1,000 patients). IPD inpatient admission yearly rates are presented for each study year in S3 Table in S1 File, with no significant trend over the study period. The mean length of IPD hospitalisation was 13.29 (95% CI 12.75–13.85) days in children aged 0–17 years from 2003–2019. The IPD length of hospitalisation per episode increased with age: 12.69 (95% CI 12.01–13.41) days in <2 years, 12.83 (95% CI 11.69–14.06) days in 2–4 years, and 15.61 (95% CI 14.29–17.01) days in 5–17 years.

The inpatient IPD cost per episode was £34,255 (95% CI 27,222–41,288) across the study period (Table 3). The eldest children (5–17 years) had the highest inpatient costs, £37,960 (95% CI 5,841–70,078), compared with <2 years [£33,192 (95% CI 27,871–38,514)] and 2–4 years [£34,034 (95% CI 21,243–46,824)]. Changes in inpatient costs per episode over the years are presented in Fig 1, and no significant changes were observed in the Mann-Kendall tests (S4

**Table 2. IPD, PP, ACP and AOM HCRU by age groups overall the study period (2003–2019).**

| | All children | <2 years | 2–4 years | 5–17 years |
|---|---|---|---|---|
| **IPD** | | | | |
| N episodes | 170 | 101 | 36 | 33 |
| Inpatient admission yearly rate per 1,000 patients (95% CI) | 0.04 (0.03–0.04) | 0.28 (0.24–0.33) | 0.04 (0.03–0.05) | 0.01 (0.01–0.01) |
| Days of hospitalisation per episode (95% CI) | 13.29 (12.75–13.85) | 12.69 (12.01–13.41) | 12.83 (11.69–14.06) | 15.61 (14.29–17.01) |
| **PP** | | | | |
| N episodes | 769 | 195 | 265 | 309 |
| GP visits yearly rate per 1,000 patients (95% CI) | 0.09 (0.09–0.10) | 0.26 (0.22–0.31) | 0.20 (0.18–0.23) | 0.05 (0.05–0.06) |
| GP antibiotic prescription yearly rate per 1,000 patients (95% CI) | 0.03 (0.03–0.03) | 0.06 (0.04–0.08) | 0.05 (0.04–0.07) | 0.02 (0.02–0.02) |
| Inpatient admission yearly rate per 1,000 patients (95% CI) | 0.02 (0.02–0.02) | 0.07 (0.05–0.10) | 0.03 (0.02–0.05) | 0.01 (0.01–0.01) |
| Days of hospitalisation per episode (95% CI) | 1.18 (1.11–1.26) | 1.07 (0.93–1.22) | 1.30 (1.17–1.45) | 1.15 (1.04–1.28) |
| **ACP** | | | | |
| N episodes | 12,142 | 3,571 | 4,358 | 4,213 |
| GP visits yearly rate per 1,000 patients (95% CI) | 1.12 (1.09–1.14) | 3.32 (3.18–3.46) | 2.34 (2.26–2.43) | 0.58 (0.56–0.60) |
| GP antibiotic prescription yearly rate per 1,000 patients (95% CI) | 0.30 (0.29–0.31) | 0.78 (0.71–0.85) | 0.60 (0.56–0.65) | 0.02 (0.02–0.02)0.18 (0.17–0.19) |
| Inpatient admission yearly rate per 1,000 patients (95% CI) | 1.24 (1.22–1.27) | 4.29 (4.14–4.46) | 2.76 (2.67–2.85) | 0.54 (0.52–0.56) |
| Days of hospitalisation per episode (95% CI) | 3.07 (3.04–3.10) | 3.46 (3.40–3.52) | 2.86 (2.81–2.91) | 2.96 (2.91–3.01) |
| **AOM** | | | | |
| N episodes | 274,008 | 53,737 | 99,641 | 120,630 |
| GP visits yearly rate per 1,000 patients (95% CI) | 38.47 (38.33–38.61) | 87.64 (86.92–88.36) | 82.67 (82.16–83.17) | 22.63 (22.50–22.76) |
| GP antibiotic prescription yearly rate per 1,000 patients (95% CI) | 35.24 (35.10–35.37) | 82.07 (81.38–82.77) | 76.00 (75.51–76.48) | 20.45 (20.33–20.57) |

ACP: all-cause pneumonia; AOM: acute otitis media; CI: confidence interval; GP: general practice; HCRU: healthcare resource utilisation; IPD: invasive pneumococcal disease; N: number; PP: pneumococcal pneumonia.

Table in S1 File). The total inpatient annual cost was £5,907,285 in children aged ≤17 years from 2003–2019 (Table 3).

## Pneumonia (PP and ACP)

In children aged 0–17 years there were 769 PP episodes and 12,142 ACP episodes in both primary care and hospital settings from 2003 to 2019. For PP and ACP, the GP visits yearly rates were 0.09 (95% CI 0.09–0.10) and 1.12 (95% CI 1.09–1.14) per 1,000 patients, respectively, across the total study period (Table 2). From 2003–2019, the decreasing trend in GP visits yearly rates was significant for both PP and ACP overall (p-value<0.001), and by age groups (for PP, p-value = 0.003 for children aged <2 and 5–17, p-value<0.001 for children aged 2–4; for ACP, p-value<0.001 for all age groups–S5 and S6 Tables in S1 File). Across the study period, the GP antibiotic prescription yearly rate was 0.03 (95% CI 0.03–0.03) for PP and 0.30 (95% CI 0.29–0.31) for ACP per 1,000 patients (Table 2).

In the inpatient setting, the inpatient admission yearly rate was 0.02 (95% CI 0.02–0.02) for PP and 1.24 (95% CI 1.22–1.27) for ACP in children aged ≤17 years over the study period (per 1,000 patients) (Table 2). The decrease observed in the inpatient admission yearly rates was significant in children aged 2–4 years (p-value = 0.005) and 5–17 years (p-value = 0.028) for PP, and in children aged <2 years (p-value<0.001) for ACP (S7 Table in S1 File for PP, and S8 Table in S1 File for ACP in hospital setting). The mean length of hospitalisations was 1.18 (95% CI 1.11–1.26) days for PP and 3.07 (95% CI 3.04–3.10) days for ACP in children aged 0–17 years (Table 2).

**Table 3. IPD, PP, ACP and AOM costs per episode by age groups overall the study period (2003–2019).**

| | All children | <2 years | 2–4 years | 5–17 years |
|---|---|---|---|---|
| **IPD** | | | | |
| Inpatient cost per episode, £ (95% CI) | 34,255 (27,222–41,288) | 33,192 (27,871–38,514) | 34,034 (21,243–46,824) | 37,960 (5,841–70,078) |
| Total inpatient annual costs, £ | 5,907,285 | 3,352,430 | 1,208,130 | 1,346,725 |
| **PP** | | | | |
| Primary care cost per episode, £ (95% CI) | 38.4 (37.0–39.7) | 35.9 (33.6–38.2) | 39.5 (37.3–41.7) | 39.1 (36.8–41.4) |
| Inpatient cost per episode, £ (95% CI) | 1,498 (1,153–1,843) | 1,338 (800–1,875) | 1,651 (929–2,374) | 1,459 (968–1,950) |
| Total cost (Primary care + Inpatient) per episode, £ (95% CI) | 1,536 (1,192–1,880) | 1,373 (837–1,910) | 1,691 (969–2,412) | 1,498 (1,008–1,988) |
| Primary care total annual costs, £ | 29,568 | 6,994 | 10,482 | 12,092 |
| Inpatient total annual costs, £ | 1,139,841 | 260,822 | 432,613 | 446,406 |
| Total annual costs (Primary care + Inpatient), £ | 1,169,409 | 267,816 | 443,095 | 458,498 |
| **ACP** | | | | |
| Primary care cost per episode, £ (95% CI) | 28.6 (28.2–29.1) | 24.7 (23.9–25.4) | 28.1 (27.4–28.8) | 32.4 (31.6–33.1) |
| Inpatient cost per episode, £ (95% CI) | 3,549 (3,405–3,693) | 4,331 (3,747–4,914) | 3,450 (3,251–3,650) | 3,316 (3,116–3,516) |
| Total cost (Primary care + Inpatient) per episode, £ (95% CI) | 3,577 (3,433–3,721) | 4,355 (3,772–4,939) | 3,478 (3,279–3,678) | 3,348 (3,148–3,548) |
| Primary care total annual costs, £ | 344,692 | 88,368 | 121,870 | 134,455 |
| Inpatient total annual costs, £ | 46,782,367 | 15,507,600 | 15,646,788 | 15,627,979 |
| Total annual costs (Primary care + Inpatient), £ | 47,127,059 | 15,595,967 | 15,768,658 | 15,762,434 |
| **AOM** | | | | |
| Primary care cost per episode, £ (95% CI) | 48.7 (48.7–48.7) | 48.8 (48.7–48.9) | 48.4 (48.3–48.5) | 49.2 (49.1–49.3) |
| Primary care total annual costs, £ | 13,484,959 | 2,647,660 | 4,857,246 | 5,980,053 |

ACP: all-cause pneumonia; AOM: acute otitis media; CI: confidence interval; IPD: invasive pneumococcal disease; PP: pneumococcal pneumonia.

From 2003–2019, the primary care costs per episode were £38.4 (95% CI 37.0–39.7) for PP and £28.6 (95% CI 28.2–29.1) for ACP. Regarding inpatient care, the costs per episode across the study period were £1,498 (95% CI 1,153–1,843), and £3,549 (95% CI 3,405–3,693) for PP and ACP, respectively (Table 3). The PP and ACP changes in costs per episode over the years are presented in Fig 1. No significant trend was observed for PP (S9 Table in S1 File). A significant decreasing trend was only observed in ACP primary care costs per episode in children overall, and when stratified by age group (p-value<0.001), and for the inpatient cost per episode in children aged <2 years (p-value = 0.036) (S10 Table in S1 File). Total PP annual costs for children aged ≤17 years were £29,568 in primary care and £1,139,841 for inpatient care. For ACP, the total primary care and inpatient annual costs were £344,692 and £46,782,367 respectively (Table 3).

## AOM

Across the study period (2003–2019), 274,008 AOM episodes were identified in primary care, with 56% of episodes occurring in children aged <5 years (Table 2). The AOM GP visits yearly rate was 38.47 (95% CI 38.33–38.61) and the AOM GP antibiotic prescription yearly rate was 35.24 (95% CI 35.10–35.37) per 1,000 patients in children overall during the study period. The AOM GP visits yearly rates were lower in children aged 5–17 years [22.63 (95% CI 22.50–22.76)], compared to both children aged <2 years [87.64 (95% CI 86.92–88.36)] and children aged 2–4 years [82.67 (95% CI 82.16–83.17)] (per 1,000 patients). A significant decreasing trend (p-value<0.001) was observed in AOM GP visits yearly rate from 2003 to 2019 in children overall and by age groups (S11 Table in S1 File).

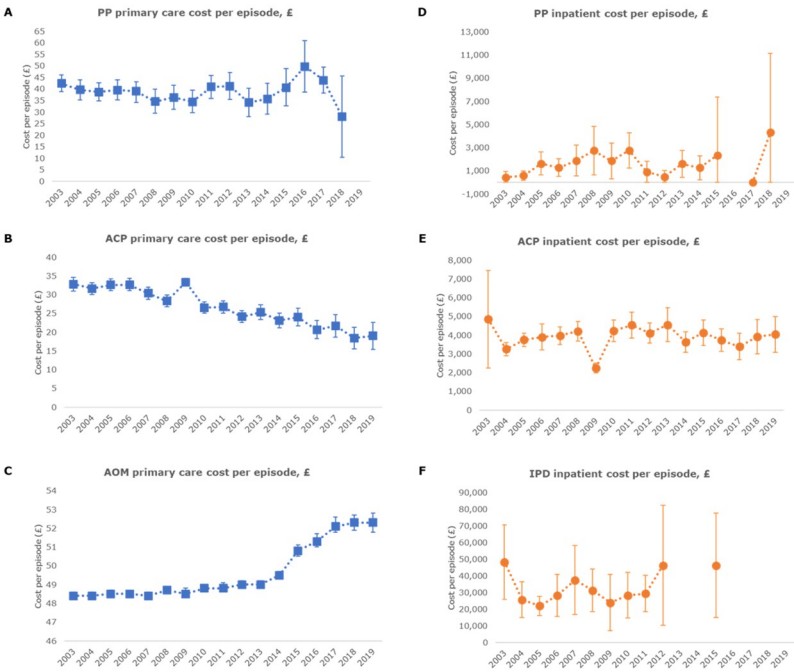

**Fig 1. PP, ACP, AOM and IPD costs per episode by study year.** PP primary care cost per episode (A), ACP primary care cost per episode (B), AOM primary care cost per episode (C), PP Inpatient cost per episode (D), ACP Inpatient cost per episode (E) and IPD Inpatient cost per episode (F). For figures D and F, where the number of episodes were less than 5, the data are not shown, in accordance with CPRD and HES-APC data protection policies. ACP: all-cause pneumonia; AOM: acute otitis media; IPD: invasive pneumococcal disease; PP: pneumococcal pneumonia.

The primary care cost per episode was £48.7 (95% CI 48.7–48.7) during the studied years (Table 3). The primary care costs per episode across the study period are presented in Fig 1. A significant increasing trend was detected in the overall population (children 0–17 years) and in children aged 2–4 years and 5–17 years (p-value<0.001) (S12 Table in S1 File). From 2003–2019 the total primary care annual cost was £13,484,959 in children aged ≤17 years (Table 3).

## Discussion

Understanding the economic burden of pneumococcal disease in children in England before higher valent PCVs are introduced is essential. This study provides HCRU and cost data in children ≤17 years in England from 2003, before any PCV vaccine was introduced, up to the late post-PCV13 period (2019). Among children in England, ACP had the highest inpatient admission yearly rate per 1,000 patients compared to IPD and PP. In contrast, IPD had the highest inpatient cost per episode followed by ACP and PP. Regarding primary care HCRU and costs, GP visits yearly rates were highest for AOM, followed by ACP, and then PP. The primary care costs per episode were highest for AOM followed by PP and ACP. By age group, the highest inpatient admission yearly rates and GP visits yearly rates were observed in children aged <2 years. No clear pattern in cost by episode by age group was observed.

From 2003 to 2019, there was no significant trend in IPD inpatient admission yearly rates nor inpatient costs. Regarding GP visits yearly rates, a significant decreasing trend was seen for PP and ACP (overall and by age groups). There was also a significant decreasing trend in the inpatient admission yearly rates for PP in children aged 2–4 years and 5–17 years, and for ACP in children <2 years. For PP costs, no significant trends in inpatient or primary care

costs were observed across the study period. In ACP a significant decrease in primary care costs overall and by age group was observed, as well as for the inpatient cost per episode in children aged <2 years. In the case of AOM, a significant decreasing trend in GP visits yearly rates were also observed (overall and by age groups). In contrast, however there was a significant increasing trend in AOM primary care costs overall and in both children aged 2–4 years and 5–17 years. This was likely driven by the increase in GP antibiotic prescription cost per episode from £7.8 (95% CI 7.7–7.9) in 2003 to £11.8 (95% CI 11.5–12.1) in 2019 (results not presented).

To our knowledge there are no prior studies that report HCRU and cost of PD in children in England using real world data. A previous study by Hu T., et al (2021) modelled the IPD health and economic burden in a hypothetical unvaccinated birth cohort after the introduction of the new PCV-15 in eight countries in Europe using a Markov model [25]. The countries included were the UK, France, Germany, Italy, Spain, Denmark, Norway and Switzerland. For the UK, cost inputs for the model came from a prior paper by Delgleize E., et al (2016), who had similarly estimated vaccine cost-effectiveness using a Markov model using NHS reference costs (2013–2014) [49]. Hu and colleagues estimated the direct medical cost per episode of meningitis and bacteraemia as €9,780 and €8,194 (2018 euros), respectively; corresponding to £8,580 and £7,189, based on 2019 average exchange rate [50]. Our IPD definition includes other manifestations of IPD, such as bacteraemic pneumonia and pericarditis, in addition to meningitis and bacteraemia. Therefore, the direct medical cost per episode of IPD estimated in the present study is considerably higher than estimated previously by Hu and colleagues [£34,255 (2019 pounds) inpatient cost per episode in the present study; see S2 Table in S1 File for Healthcare resource reference units costs per IPD manifestation].

Given the differences in healthcare systems and social services among countries, together with variation in age structures, it is difficult to compare economic burden studies from diverse countries, especially when comparing high-income countries, such as England, to low-income countries. Studies elsewhere in Europe reporting HCRU and costs of IPD in children are also scarce. Brotons P., et al (2013) conducted an observational study in 2001–2011, including 135 children aged <18 years with culture-proved IPD admitted to a referral hospital in Spain [51]. The median hospital cost of IPD was estimated at €4,533 (95% CI 4,078–5,435) (£3,977, based on 2019 average exchange rate [50]), and the median length of stay in hospital of 11.0 (95% CI 10.6–13.0) days. This is an alignment with the length of hospitalisation reported in our study, 13.29 (95% CI 12.75–13.85) days. More recently, Amicizia D., et al (2022) estimated the cost of emergency department (ED) visits and hospitalisations associated with IPD, ACP, and AOM in children <15 years of age in the Liguria region of Italy between 2012 and 2018, using the Liguria Region Administrative Health Databases and the Ligurian Chronic Condition Data Warehouse databases. In total, 9,750 ACP episodes (including 287 PP episodes), 17,040 AOM episodes and 878 IPD episodes were identified in children in Liguria. From 2012–2018, the median IPD cost per ED visit or hospitalisation was €891 (£782 based on 2019 average exchange rate [50]). Due to the low number of IPD episodes identified in our study, the confidence intervals for inpatient cost per episode are very wide [e.g., children aged 5–17 years - £37,960 (95% CI 5,841.3–70,078)], making comparisons difficult and imprecise.

Focusing on pneumonia, Nair H., et al evaluated the effect of PCV7 and PCV13 on ACP hospitalisation rates, using hospital records in Scotland [52]. The ACP hospitalization rates decreased by about 30% in children aged <2 years in the post-PCV13 period (2010–2012) compared to the pre-PCV period (2000–2005) [52]. In our study we also observed a decrease across the study period in the inpatient admission yearly rate in children aged <2 years, from 4.67 (95% CI 3.99–5.43) in 2003 to 2.55 (95% CI 1.70–3.66) in 2019, per 1,000 patients (p-value <0.001). Previous studies reporting healthcare costs for ACP in the UK are limited. A

previous study by Guest J.F. and Morris A. (1997) reported direct annual healthcare costs associated with community-acquired pneumonia during 1992/1993 [53]. The study was conducted in HES, and supplemented by telephone surveys, and no age restrictions were applied. The average cost for managing pneumonia in the community was reported to be £100 per episode, compared to £1,700–5,100 when the patient was hospitalised, depending on the length of hospitalisation. The ACP primary care cost per episode was lower in our study [£28.6 (95% CI 28.2–29.1)] but the inpatient cost per hospital episode was within the range mentioned [£3,549 (95% CI 3,405–3,693)]. Several years have passed since Guest´s publication [53], and more recent studies assessing the direct cost per pneumonia episode in children in England are lacking. In a recent Italian study, Amicizia D., et al reported the median cost per ED visit or hospitalisation from 2012–2018 for PP (€3,323), and ACP (€1,850) [54], corresponding to £2,915 and £1,623 based on 2019 average exchange rate [50]. This contrasts with our study, where we found higher inpatient costs per episode in ACP compared to PP from 2003–2019, £3,549 (95% CI 3,405–3,693) for ACP versus £1,498 (95% CI 1,153–1,843) for PP. This is likely explained by the differences in the coding of PP and ACP. For ACP, our study had a broader definition, including any unknown and known pathogens, while in the Italian study, only "Bronchopneumonia, organism unspecified" and "Pneumonia, organism unspecified" diagnosis codes were included. Furthermore, the number of PP episodes were two-fold higher in our study (769 PP episodes).

AOM is one of the most common diseases in early infancy and childhood. Healthcare costs for AOM have previously been reported for the UK. In a survey study of parents of children <5 years from seven European countries including the UK, the total cost per AOM episode was €752.49 [55] (£660.16 based on 2019 average exchange rate [50]). This figure included indirect costs such as productivity losses and is therefore higher than the AOM cost per episode we report in the present study. A recent study conducted in the Netherlands by van Uum R.T. et al (2021) [56], explored the societal cost of AOM in 224 children aged six months to 10 years. This cluster randomised controlled trial compared the effectiveness of an intervention aimed at educating GPs about pain management in AOM compared to usual care between 2015 and 2018. The mean total health care costs were €77.60 per child (£68.08 based on 2019 average exchange rate [50]). The largest contributors to these costs were GP consultations and hospital admissions, at €49.80, SD 1.77 (£43.69, based on 2019 average exchange rate [50]) and €10.16, SD 10.16 (£8.91, based on 2019 average exchange rate [50]) per patient, respectively. The primary care cost estimate is in alignment with our study [£48.7 (95% CI 48.7–48.7)].

The results of our study should be interpreted in the context of the following limitations. Firstly, there was a reduction in the size of the study population in CPRD-Gold from 2015 onwards. The migration of GP practices from one GP software to another explains this size reduction [57]. In spite of this reduction in study population size, CPRD-Gold continues to be representative of the UK population [31]. Secondly, the number of PP and IPD episodes in the present study was small resulting in wide confidence intervals, so the assessment of the trends over time may not be precise and should be interpreted with caution. Significant variation in incidence of PP versus ACP has previously been reported; likely due to underreporting of PP [58]. This underreporting of PP is not surprising, as in clinical practice, especially in primary care, initial pneumonia diagnosis is typically made on clinical judgment without radiological confirmation or knowledge of the causative organism [59]. This may have led to underestimation of the true PD economic burden. We therefore included a wider pneumonia definition, ACP, to capture all pneumonia episodes caused by any organisms to estimate the economic burden of PD. It has been suggested that 30% of ACP is PP [60]. In the present study, PP comprised 7% of ACP overall. This estimate is not dissimilar from serological studies in the US,

suggesting that *S. Pneumoniae* is reponsible for 4% of hospitalised community-acquired pneumonia in children [34].

Furthermore there is a potential risk of misclassification bias due to coding inaccuracies, as medical conditions were identified based on existing records. However, previous validation studies suggest that ICD-based claim codes have good sensitivity and specificity for a diagnosis of pneumonia [61, 62]. To minimise misclassification bias, careful and thorough evaluation of pneumococcal-specific and unspecified diagnoses used to identify pneumococcal-related infections as well as rules for defining episodes was performed. While these steps would not have prevented coding errors or omissions, it did reduce the risk of misclassification, either due to lack of specificity or sensitivity of the diagnosis codes used to identify pneumococcal-related infections, or the episode definitions that are not reflective of the typical duration of illness.

There were also limitations related to the data that particularly affected the calculation of HCRU and costs. Costs were calculated based on HRG groups over a long follow-up period where different accounting systems were used. This was clear in the variability in the mean case cost per day by disease type. To mitigate against this, the most updated HRG tariffs for each HRG code were identified and used for the calculations of cost. In addition, indirect costs were not estimated, so the total direct costs are likely to be an underestimate.

These data demonstrate that while HCRU may have decreased in primary care for PD, the economic burden of PD and ACP among children in inpatient care continues to be high despite the introduction of PCV13 in England. Estimating the economic burden of PD is important when considering the development and introduction of novel higher-valent vaccines to reduce residual disease burden. The increased prevalence and pathogenicity of non-vaccine serotypes, and persistence of certain vaccine serotypes (primarily serotypes 3 and 19A [23]), pose new challenges to the reduction in the burden of PD [63]. A recent study conducted by Hanquet G., et al (2022) using surveillance data from 13 SpIDnet sites from 10 European countries [64], estimated the overall effect of the childhood PCV10/PCV13 program by comparing IPD incidence before and after vaccine introduction. In children aged <5 years, the IPD caused by non-PCV13 serotypes increased gradually until 2018, when it exceeded the PCV7 period incidence by 111%. This increase was likely due to a combination of vaccine-induced serotype replacement along with disease and secular trends in individual serotypes. Similar trends have been reported for other PD outcomes [65–69]. While universal immunisation programs and better management of complications will contribute to reducing the cost burden, these findings highlight the need for novel vaccines to reduce the burden of PD in children.

## Conclusions

HCRU and cost per episode was highest for IPD in inpatient care, and highest for AOM in primary care in children aged ≤17 years in England from 2003 to 2019. In the primary care setting, in children overall a significant decreasing trend was observed in HCRU for PP, ACP and AOM. Across the study period, the primary care cost decreased for ACP while it increased for AOM. In the inpatient setting, no significant trends were observed in inpatient admission yearly rates for IPD, PP or ACP. Inpatient costs did not vary significantly across the study period for IPD, PP or ACP. The economic burden of IPD, PP, ACP, and AOM remains substantial in England.

## Supporting information

**S1 File.**
(DOCX)

## Acknowledgments

This study is based in part on data from the CPRD and HES APC databases, obtained under license from the UK Medicines and Healthcare products Regulatory Agency. The data were provided by patients and collected by the National Health Service as part of their care and support.

## Author Contributions

**Conceptualization:** Salini Mohanty, Bélène Podmore, Ana Cuñado Moral, Ian Matthews, Eric Sarpong, Agueda Azpeitia.

**Data curation:** Agueda Azpeitia.

**Formal analysis:** Agueda Azpeitia.

**Funding acquisition:** Salini Mohanty, Ian Matthews.

**Investigation:** Salini Mohanty, Bélène Podmore, Ana Cuñado Moral, Ian Matthews, Eric Sarpong, Agueda Azpeitia, Nawab Qizilbash.

**Methodology:** Salini Mohanty, Bélène Podmore, Ana Cuñado Moral, Ian Matthews, Eric Sarpong, Agueda Azpeitia, Nawab Qizilbash.

**Project administration:** Salini Mohanty, Bélène Podmore.

**Software:** Agueda Azpeitia.

**Supervision:** Salini Mohanty, Bélène Podmore, Eric Sarpong, Nawab Qizilbash.

**Validation:** Agueda Azpeitia.

**Visualization:** Agueda Azpeitia.

**Writing – original draft:** Bélène Podmore, Ana Cuñado Moral.

**Writing – review & editing:** Salini Mohanty, Bélène Podmore, Ana Cuñado Moral, Ian Matthews, Eric Sarpong, Agueda Azpeitia, Nawab Qizilbash.

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
