## [Decision Letter · Decision Letter 0]

7 Dec 2022

PONE-D-22-26429Healthcare resource utilisation and cost of pneumococcal disease from 2003 to 2019 in children ≤17 years in EnglandPLOS ONE

Dear Dr. Salini Mohanty,

Thank you for submitting your manuscript to PLOS ONE. After careful consideration, we feel that it has merit but does not fully meet PLOS ONE’s publication criteria as it currently stands. Therefore, we invite you to submit a revised version of the manuscript that addresses the points raised during the review process.

 Please submit your revised manuscript by Jan 21 2023 11:59PM. If you will need more time than this to complete your revisions, please reply to this message or contact the journal office at plosone@plos.org. Please include the following items when submitting your revised manuscript:A rebuttal letter that responds to each point raised by the academic editor and reviewer(s). You should upload this letter as a separate file labeled 'Response to Reviewers'.A marked-up copy of your manuscript that highlights changes made to the original version. You should upload this as a separate file labeled 'Revised Manuscript with Track Changes'.An unmarked version of your revised paper without tracked changes. You should upload this as a separate file labeled 'Manuscript'.If applicable, we recommend that you deposit your laboratory protocols in protocols.io to enhance the reproducibility of your results. Protocols.io assigns your protocol its own identifier (DOI) so that it can be cited independently in the future. For instructions see: https://journals.plos.org/plosone/s/submission-guidelines#loc-laboratory-protocols. Additionally, PLOS ONE offers an option for publishing peer-reviewed Lab Protocol articles, which describe protocols hosted on protocols.io. Read more information on sharing protocols at https://plos.org/protocols?utm_medium=editorial-email&utm_source=authorletters&utm_campaign=protocols.

We look forward to receiving your revised manuscript.

Kind regards,

Rahul Garg, MD

Academic Editor

PLOS ONE

Journal Requirements:

2. Thank you for providing the following Funding Statement: 

“I have read the journal’s policy and the authors of this manuscript have the following competing interests: Salini Mohanty, Ian Matthews, and Eric Sarpong are employees of Merck Sharp & Dohme LLC, a subsidiary of Merck & Co., Inc., Rahway, NJ, USA and may own stock/stock options in Merck & Co., Inc., Rahway, NJ, USA. Bélène Podmore, Ana Cuñado, Agueda Azpeitia, and Nawab Qizilbash are employees of OXON Epidemiology Ltd, Epidemiology & Statistics, Madrid, Spain, an independent contract research organization, that received funding from Merck Sharp & Dohme LLC, a subsidiary of Merck & Co., Inc., Rahway, NJ, USA to design and conduct this study.”

We note that one or more of the authors is affiliated with the funding organization, indicating the funder may have had some role in the design, data collection, analysis or preparation of your manuscript for publication; in other words, the funder played an indirect role through the participation of the co-authors.

If the funding organization did not play a role in the study design, data collection and analysis, decision to publish, or preparation of the manuscript and only provided financial support in the form of authors' salaries and/or research materials, please review your statements relating to the author contributions, and ensure you have specifically and accurately indicated the role(s) that these authors had in your study in the Author Contributions section of the online submission form. Please make any necessary amendments directly within this section of the online submission form.  Please also update your Funding Statement to include the following statement: “The funder provided support in the form of salaries for authors [insert relevant initials], but did not have any additional role in the study design, data collection and analysis, decision to publish, or preparation of the manuscript. The specific roles of these authors are articulated in the ‘author contributions’ section.”

If the funding organization did have an additional role, please state and explain that role within your Funding Statement.

Please also provide an updated Competing Interests Statement declaring this commercial affiliation along with any other relevant declarations relating to employment, consultancy, patents, products in development, or marketed products, etc. 

3. PLOS requires an ORCID iD for the corresponding author in Editorial Manager on papers submitted after December 6th, 2016. Please ensure that you have an PLOS requires an ORCID iD for the corresponding author in Editorial Manager on papers submitted after December 6th, 2016. Please ensure that you have an ORCID iD and that it is validated in Editorial Manager. To do this, go to ‘Update my Information’ (in the upper left-hand corner of the main menu), and click on the Fetch/Validate link next to the ORCID field. This will take you to the ORCID site and allow you to create a new iD or authenticate a pre-existing iD in Editorial Manager. Please see the following video for instructions on linking an ORCID iD to your Editorial Manager account: https://www.youtube.com/watch?v=_xcclfuvtxQORCID iD and that it is validated in Editorial Manager. To do this, go to ‘Update my Information’ (in the upper left-hand corner of the main menu), and click on the Fetch/Validate link next to the ORCID field. This will take you to the ORCID site and allow you to create a new iD or authenticate a pre-existing iD in Editorial Manager. Please see the following video for instructions on linking an ORCID iD to your Editorial Manager account: https://www.youtube.com/watch?v=_xcclfuvtxQ.

Additional Editor Comments (if provided):

The manuscript describes an estimate of the healthcare resource utilisation (HCRU) and costs associated with pneumococcal disease (PD) in children aged ≤17 years in England from 2003-2019.

Episodes of invasive pneumococcal disease (IPD) were identified in hospitals in England, pneumococcal pneumonia (PP) and all-cause pneumonia (ACP) episodes were identified both in primary care as well as in hospitals, and acute otitis media (AOM) episodes were identified in primary care.

A respectable number of more than 1.5 million children were followed in the study period that was from 2003 to 2019. This period therefore includes a few years before introduction of any PCV, a few years after introduction of PCV 7 in 2006 and PCV 13 from 2010.

The authors declare conflicts of interest as some of the authors are employees of MSD and others of OXON, a company receiving funding from MSD. This is important as MSD will soon/already launch a 15 valent pneumococcal conjugate vaccine, obviously in competition with Pfizer´s 13 valent and GlaxoSmithKline 10 valent vaccines, both companies launching or preparing to launch higher valent vaccines. The authors do mention PCV 13 a few times in the paper but never PCV 10.

The paper is well written and carries interesting information. The paper needs some minor revision.

Reviewers' comments:

Reviewer's Responses to Questions

**Comments to the Author**

1. Is the manuscript technically sound, and do the data support the conclusions?

Reviewer #1: Yes

Reviewer #2: Yes

2. Has the statistical analysis been performed appropriately and rigorously? 

Reviewer #1: Yes

Reviewer #2: Yes

3. Have the authors made all data underlying the findings in their manuscript fully available?

Reviewer #1: Yes

Reviewer #2: Yes

4. Is the manuscript presented in an intelligible fashion and written in standard English?

Reviewer #1: Yes

Reviewer #2: Yes

5. Review Comments to the Author

Reviewer #1: The study aimed to estimate healthcare resource utilisation (HCRU) and costs associated with pneumococcal disease (PD) in children aged ≤17 years in England from 2003- 2019.

The manuscript is well written, clear and precise.

Reviewer #2: Healthcare resource utilisation and cost of pneumococcal disease from 2003 to 2019 in children ≤17 years in England

PloS One

Manuscript Number: PONE-D-22-26429

Article Type: Research Article

The manuscript describes an estimate of the healthcare resource utilisation (HCRU) and costs associated with pneumococcal disease (PD) in children aged ≤17 years in England from 2003-2019. This is a comprehensive collection of data using various health-care related data banks in England. The authors evaluated the cost of pneumococcal diseases, i.e. invasive pneumococcal disease (IPD) pneumococcal pneumonia (PP), all-cause pneumonia (ACP) and acute otitis media (AOM). The last two obviously not always caused by pneumococcus.

Episodes of invasive pneumococcal disease (IPD) were identified in hospitals in England, pneumococcal pneumonia (PP) and all-cause pneumonia (ACP) episodes were identified both in primary care as well as in hospitals, and acute otitis media (AOM) episodes were identified in primary care.

A respectable number of more than 1.5 million children were followed in the study period that was from 2003 to 2019. This period therefore includes a few years before introduction of any PCV, a few years after introduction of PCV 7 in 2006 and PCV 13 from 2010.

1. The authors declare conflicts of interest as some of the authors are employees of MSD and others of OXON, a company receiving funding from MSD. This is important as MSD will soon/already launch a 15 valent pneumococcal conjugate vaccine, obviously in competition with Pfizer´s 13 valent and GlaxoSmithKline 10 valent vaccines, both companies launching or preparing to launch higher valent vaccines. The authors do mention PCV 13 a few times in the paper but never PCV 10.

The paper is well written and carries interesting information. I only have a few minor comments or suggestions.

2. The authors might want to emphasise that the structure of the study, i.e. evaluating the cost of the healthcare resource utilisation, only includes a part of the total society cost. In one of publications by Eythorsson E (PLoS One. 2021, PMID: 33831049) the indirect cost was also substantial. There are more similar publications. This indicates that a possible decrease in HCRU may underestimates the total cost reduction after initiating PCV. Having sad that, it is briefly mentioned that there are huge differences between countries in terms of “clinical practices, cultural attitudes, health insurance initiatives” and several other factors.

3. There are a few limitations to the study, most of them addressed in the discussion. The authors may want to emphasise more clearly that the study, carried out in England, reflects the situation in a high-income country. This health-economic evaluations would obviously be very much different in low-income countries.

4. I suggest the authors state more clearly that “all cause pneumonia” and “acute otitis media” are often not caused by Streptococcus pneumonia. In children, these diseases may quite often be caused by viruses or other bacteria. It does not contaminate the results in this cohort but should be expressed more clearly.

5. In contrast, however there was a significant increasing trend in AOM primary care costs overall and in both children aged 2-4 years and 5-17 years. This was driven by the increase in GP antibiotic prescription costs (results not presented) “. Maybe not a part of this study, but in my mind, this certainly needs more explanation! Are GPs in England prescribing more antibiotics for AOM after the introduction of PCV 7 and later PCV 13? This is in contrast with some other studies on the effect of PCV´s.

6. The authors claim that “To our knowledge there are no prior studies that report HCRU and cost of PD in children in England using real world data.” And later they state that “There are no recent studies reporting healthcare costs for ACP in the UK.” This may very well be true for England and/or UK, but it would surprise me if more detailed health-economic studies on PCV’s are not available. I would encourage the authors to include other health-economic studies (not necessarily only HCRU) in their discussion. I do understand that the aim of this current study was to only evaluate the HCRU. However, the picture may be bigger, and it would be of interest to include that in the discussion – not only for AOM.

7. My main comment regards the following conclusion: “In children overall a significant decreasing trend was observed in HCRU in primary care for PP, ACP and AOM. No significant trends were observed in inpatient admission yearly rates for IPD, PP or ACP. Across the study period, the primary care cost of ACP decreased while increased for AOM. Inpatient costs did not vary significantly across the study period for IPD, PP or ACP.”

To me, there are discrepancies in these conclusions, and I still don’t quite understand them. The authors may want to explain this somewhat better in the discussion section.

8. How come that HCRU in primary care was decreasing for PP, ACP and AOM whereas no trend was observed on inpatient HCRU for IPD, PP and ACP?

9. Why did the cost of ACP in primary care decrease while it increased for AOM?

10. Wouldn’t it have been logical to expect increasing cost to decrease over the study period for IPD, PP and ACP?

11. Finally, the authors obviously need to proof-read the manuscript. “Error! Reference source not found” on several places does not reflect meticulous proof-reading before submitting.

6. PLOS authors have the option to publish the peer review history of their article (what does this mean?). If published, this will include your full peer review and any attached files.

Reviewer #1: No

Reviewer #2: No

---

## [Author Response · Author response to Decision Letter 0]

31 Jan 2023

Point-by-point response to reviewers and editors

Line numbers correspond to the revised clean version of the manuscript with tracked changes all accepted.

Responses dated 7 December 2022

Journal Requirements: 

Comment 1:

Author response: Thank you for your comment, we have reviewed the formatting along the manuscript and supporting information documents.

Comment 2: 2. Thank you for providing the following Funding Statement: 

“I have read the journal’s policy and the authors of this manuscript have the following competing interests: Salini Mohanty, Ian Matthews, and Eric Sarpong are employees of Merck Sharp & Dohme LLC, a subsidiary of Merck & Co., Inc., Rahway, NJ, USA and may own stock/stock options in Merck & Co., Inc., Rahway, NJ, USA. Bélène Podmore, Ana Cuñado, Agueda Azpeitia, and Nawab Qizilbash are employees of OXON Epidemiology Ltd, Epidemiology & Statistics, Madrid, Spain, an independent contract research organization, that received funding from Merck Sharp & Dohme LLC, a subsidiary of Merck & Co., Inc., Rahway, NJ, USA to design and conduct this study.”

We note that one or more of the authors is affiliated with the funding organization, indicating the funder may have had some role in the design, data collection, analysis or preparation of your manuscript for publication; in other words, the funder played an indirect role through the participation of the co-authors.

If the funding organization did not play a role in the study design, data collection and analysis, decision to publish, or preparation of the manuscript and only provided financial support in the form of authors' salaries and/or research materials, please review your statements relating to the author contributions, and ensure you have specifically and accurately indicated the role(s) that these authors had in your study in the Author Contributions section of the online submission form. Please make any necessary amendments directly within this section of the online submission form. Please also update your Funding Statement to include the following statement: “The funder provided support in the form of salaries for authors [insert relevant initials], but did not have any additional role in the study design, data collection and analysis, decision to publish, or preparation of the manuscript. The specific roles of these authors are articulated in the ‘author contributions’ section.”

If the funding organization did have an additional role, please state and explain that role within your Funding Statement.

Author response: Thank you for your comment, the funding statement has been updated accordingly.

Comment 3: Please also provide an updated Competing Interests Statement declaring this commercial affiliation along with any other relevant declarations relating to employment, consultancy, patents, products in development, or marketed products, etc. 

Author response: Thank you, we have updated the Competing Interests Statements accordingly.

Comment 4: 3. PLOS requires an ORCID iD for the corresponding author in Editorial Manager on papers submitted after December 6th, 2016. Please ensure that you have an PLOS requires an ORCID iD for the corresponding author in Editorial Manager on papers submitted after December 6th, 2016. Please ensure that you have an ORCID iD and that it is validated in Editorial Manager. To do this, go to ‘Update my Information’ (in the upper left-hand corner of the main menu), and click on the Fetch/Validate link next to the ORCID field. This will take you to the ORCID site and allow you to create a new iD or authenticate a pre-existing iD in Editorial Manager. Please see the following video for instructions on linking an ORCID iD to your Editorial Manager account: 

https://www.youtube.com/watch?v=_xcclfuvtxQORCID iD and that it is validated in Editorial Manager. To do this, go to ‘Update my Information’ (in the upper left-hand corner of the main menu), and click on the Fetch/Validate link next to the ORCID field. This will take you to the ORCID site and allow you to create a new iD or authenticate a pre-existing iD in Editorial Manager. Please see the following video for instructions on linking an ORCID iD to your Editorial Manager account: https://www.youtube.com/watch?v=_xcclfuvtxQ.

Author response: Thank you, we have updated accordingly the ORCID iD on the Editorial Manager Platform.

Comment 5: 4. Your ethics statement should only appear in the Methods section of your manuscript. If your ethics statement is written in any section besides the Methods, please move it to the Methods section and delete it from any other section. Please ensure that your ethics statement is included in your manuscript, as the ethics statement entered into the online submission form will not be published alongside your manuscript.

Author response: Thank you, the “ethics statement” section is now included only under the “Methods” section. 

Comment 6: 5. Please review your reference list to ensure that it is complete and correct. If you have cited papers that have been retracted, please include the rationale for doing so in the manuscript text, or remove these references and replace them with relevant current references. Any changes to the reference list should be mentioned in the rebuttal letter that accompanies your revised manuscript. If you need to cite a retracted article, indicate the article’s retracted status in the References list and also include a citation and full reference for the retraction notice.

Author response: Thank you for your comment, we have reviewed the references included in our manuscript.

Additional Editor Comments (if provided):

Comment 7: The manuscript describes an estimate of the healthcare resource utilisation (HCRU) and costs associated with pneumococcal disease (PD) in children aged ≤17 years in England from 2003-2019.

Episodes of invasive pneumococcal disease (IPD) were identified in hospitals in England, pneumococcal pneumonia (PP) and all-cause pneumonia (ACP) episodes were identified both in primary care as well as in hospitals, and acute otitis media (AOM) episodes were identified in primary care.

A respectable number of more than 1.5 million children were followed in the study period that was from 2003 to 2019. This period therefore includes a few years before introduction of any PCV, a few years after introduction of PCV 7 in 2006 and PCV 13 from 2010.

Author response: Thank you for taking the time to review the manuscript.

Comment 8: The authors declare conflicts of interest as some of the authors are employees of MSD and others of OXON, a company receiving funding from MSD. This is important as MSD will soon/already launch a 15 valent pneumococcal conjugate vaccine, obviously in competition with Pfizer´s 13 valent and GlaxoSmithKline 10 valent vaccines, both companies launching or preparing to launch higher valent vaccines. The authors do mention PCV 13 a few times in the paper but never PCV 10.

Author response: Thank you for your comment. The PCV vaccines schedules and changes in the UK have been mentioned in the introduction section. In 2010, a pneumococcal conjugate vaccine protecting against 13 types of pneumococcal bacteria (PCV13) replaced PCV7 (References: Public Health England (2019). While PCV10 was license in the UK it is not used in the UK immunisation programme. Changes to the infant pneumococcal conjugate vaccine schedule. Available from: https://assets.publishing.service.gov.uk/government/uploads/system/uploads/attachment_data/file/854153/Letter_NHSEI_PHE_PCV13_schedule_change.pdf and Vaccine Knowledge Project. PCV (Pneumococcal Conjugate Vaccine). Available from: https://vk.ovg.ox.ac.uk/vk/pcv). 

Comment 9: The paper is well written and carries interesting information. The paper needs some minor revision.

Author response: Thank you so much for your comments and for taking the time to review our manuscript. We have responded to each of your comments below in detail and made amendments to the manuscript where appropriate.

Reviewers' comments: Reviewer's Responses to Questions 

Reviewer 1

Comment 1:

Reviewer #1: The study aimed to estimate healthcare resource utilisation (HCRU) and costs associated with pneumococcal disease (PD) in children aged ≤17 years in England from 2003- 2019.

The manuscript is well written, clear and precise.

Author response: Thank you for taking the time to review the manuscript. We appreciate your feedback.

Reviewer 2

Comment 1:

Reviewer #2: Healthcare resource utilisation and cost of pneumococcal disease from 2003 to 2019 in children ≤17 years in England

PloS One

Manuscript Number: PONE-D-22-26429

Article Type: Research Article

The manuscript describes an estimate of the healthcare resource utilisation (HCRU) and costs associated with pneumococcal disease (PD) in children aged ≤17 years in England from 2003-2019. This is a comprehensive collection of data using various health-care related data banks in England. The authors evaluated the cost of pneumococcal diseases, i.e. invasive pneumococcal disease (IPD) pneumococcal pneumonia (PP), all-cause pneumonia (ACP) and acute otitis media (AOM). The last two obviously not always caused by pneumococcus.

Episodes of invasive pneumococcal disease (IPD) were identified in hospitals in England, pneumococcal pneumonia (PP) and all-cause pneumonia (ACP) episodes were identified both in primary care as well as in hospitals, and acute otitis media (AOM) episodes were identified in primary care.

A respectable number of more than 1.5 million children were followed in the study period that was from 2003 to 2019. This period therefore includes a few years before introduction of any PCV, a few years after introduction of PCV 7 in 2006 and PCV 13 from 2010.

Author response: Thank you for taking the time to review the manuscript and giving your detailed feedback.

Comment 2: 1. The authors declare conflicts of interest as some of the authors are employees of MSD and others of OXON, a company receiving funding from MSD. This is important as MSD will soon/already launch a 15 valent pneumococcal conjugate vaccine, obviously in competition with Pfizer´s 13 valent and GlaxoSmithKline 10 valent vaccines, both companies launching or preparing to launch higher valent vaccines. The authors do mention PCV 13 a few times in the paper but never PCV 10.

The paper is well written and carries interesting information. I only have a few minor comments or suggestions.

Author response: Thank you for your feedback. Regarding PCV information, the PCV vaccines schedules and changes in the UK have been mentioned in the introduction section. In 2010, a pneumococcal conjugate vaccine protecting against 13 types of pneumococcal bacteria (PCV13) replaced PCV7 (References: Public Health England (2019). While PCV10 was license in the UK it is not used in the UK immunisation programme. Changes to the infant pneumococcal conjugate vaccine schedule. Available from: https://assets.publishing.service.gov.uk/government/uploads/system/uploads/attachment_data/file/854153/Letter_NHSEI_PHE_PCV13_schedule_change.pdf and Vaccine Knowledge Project. PCV (Pneumococcal Conjugate Vaccine). Available from: https://vk.ovg.ox.ac.uk/vk/pcv).

Comment 3: 2. The authors might want to emphasise that the structure of the study, i.e. evaluating the cost of the healthcare resource utilisation, only includes a part of the total society cost. In one of publications by Eythorsson E (PLoS One. 2021, PMID: 33831049) the indirect cost was also substantial. There are more similar publications. This indicates that a possible decrease in HCRU may underestimates the total cost reduction after initiating PCV. Having sad that, it is briefly mentioned that there are huge differences between countries in terms of “clinical practices, cultural attitudes, health insurance initiatives” and several other factors.

Author response: Thank you for your comment and for sharing this interesting cost-effectiveness study that highlights the importance of PCVs on children. We have added in the discussion that this is likely to be an underestimate as indirect costs have not been taken into account (see line 355). 

Comment 4: 3. There are a few limitations to the study, most of them addressed in the discussion. The authors may want to emphasise more clearly that the study, carried out in England, reflects the situation in a high-income country. This health-economic evaluations would obviously be very much different in low-income countries.

Author response: Thank you for your comment, we have emphasised this in the discussion section, see line 284.

Comment 5: 4. I suggest the authors state more clearly that “all cause pneumonia” and “acute otitis media” are often not caused by Streptococcus pneumonia. In children, these diseases may quite often be caused by viruses or other bacteria. It does not contaminate the results in this cohort but should be expressed more clearly.

Author response: Thank you for your comment. We have added additional information to the background in order to provide detail on S. pneumoniae causing AOM and pneumonia (see line 49 and 53).

Comment 6: 5. In contrast, however there was a significant increasing trend in AOM primary care costs overall and in both children aged 2-4 years and 5-17 years. This was driven by the increase in GP antibiotic prescription costs (results not presented) “. Maybe not a part of this study, but in my mind, this certainly needs more explanation! Are GPs in England prescribing more antibiotics for AOM after the introduction of PCV 7 and later PCV 13? This is in contrast with some other studies on the effect of PCV´s.

Author response: Thank you for your comment. The GP prescription cost per episode increased from £7.8 (95% CI 7.7-7.9) in 2003 to £11.8 (95% CI 11.5-12.1) in 2019, while the GP visit cost per episode has not varied significations over the study period, from £40.7 (95% CI 40.5-40.8) to £40.5 (95% CI 40.2-40.8). Discussion edited adding more detail, including the GP prescription costs for AOM (see line 268).

Comment 7: 6. The authors claim that “To our knowledge there are no prior studies that report HCRU and cost of PD in children in England using real world data.” And later they state that “There are no recent studies reporting healthcare costs for ACP in the UK.” This may very well be true for England and/or UK, but it would surprise me if more detailed health-economic studies on PCV’s are not available. I would encourage the authors to include other health-economic studies (not necessarily only HCRU) in their discussion. I do understand that the aim of this current study was to only evaluate the HCRU. However, the picture may be bigger, and it would be of interest to include that in the discussion – not only for AOM.

Author response: Thank you for your comment. We have carefully reviewed the literature related to health economics in the UK in children. Delgleize et al (DOI: 10.1136/bmjopen-2015-010776) compared PHiD-CV vaccine and the PCV-13 vaccine using a Markov model to evaluate the cost-effectiveness analysis of routine pneumococcal vaccination in the UK. This study is discussed in the discussion section (see line 274). 

A number of other studies (Van hoek (2012) et al. (DOI: 10.1016/j.vaccine.2012.10.017)., McIntosh et al (2005) (DOI: 10.1016/j.vaccine.2004.08.05) and Rozenbaum et al (2012) (doi: 10.1136/bmj.e6879) have also investigated the cost-effectiveness of PCVs in the UK but do not report trends across PCV periods to allow for comparison with our study. 

The study conducted by Nair et al (https://doi.org/10.1186/s12879-016-1693-x) in children and adults in Scotland evaluating pneumonia hospitalization rates across the PCV periods has been added to our discussion of the trends in HCRU for ACP (see line 298).

Comment 8: 7. My main comment regards the following conclusion: “In children overall a significant decreasing trend was observed in HCRU in primary care for PP, ACP and AOM. No significant trends were observed in inpatient admission yearly rates for IPD, PP or ACP. Across the study period, the primary care cost of ACP decreased while increased for AOM. Inpatient costs did not vary significantly across the study period for IPD, PP or ACP.”

To me, there are discrepancies in these conclusions, and I still don’t quite understand them. The authors may want to explain this somewhat better in the discussion section.

Author response: Thank you for your comment, for clarity we have rephrased the conclusions and separated out the conclusions relating to primary care and inpatients settings (see line 374).

Comment 9: 8. How come that HCRU in primary care was decreasing for PP, ACP and AOM whereas no trend was observed on inpatient HCRU for IPD, PP and ACP?

Author response: Thank you for your comment. Different outcomes were evaluated in each setting. Regarding IPD and PP, there were a low number of episodes, and in some years, there were <5 episodes. Where the number of hospital admissions/GP visits were less than 5 the data are not reported, in accordance with CPRD and HES-APC data protection policies. As there are years with gaps (due to the low number of episodes), this could have affected the trend analysis. Regarding inpatient ACP, the number of hospital admissions was enough to perform the inpatient admission yearly rates Mann-Kendall tests, and these were not significant.

Comment 10: 9. Why did the cost of ACP in primary care decrease while it increased for AOM?

Author response: Thank you for your comment. The primary care costs included GP visit costs and GP prescription costs. The trend in primary care costs for ACP and AOM were different, and this might be due to the increase in AOM prescription costs per episode (from £7.8 (95% CI 7.7-7.9) in 2003 to £11.8 (95% CI 11.5-12.1 in 2019), detailed on this GP prescription cost increase has been added to the discussion. A prescription cost increase was not seen in ACP GP prescriptions, varying from £1.8 (95% CI 1.4-2.1) to £1.6 (95% CI 0.9-2.4) in 2019. Focusing on GP visit cost per episode, a more marked decrease was seen in ACP compared to AOM. ACP GP visit cost per episode decreased from £31.0 (95% CI 29.3-32.7) in 2003 to £17.3 (95% CI 14.1-20.6) in 2019). AOM visit cost per episode did not vary significantly from £40.7 (95% CI 40.5, 40.8) in 2003 to £40.5 (95% CI 40.2, 40.8) in 2019. Amicizia, et al (https://doi.org/10.1080/21645515.2022.2082205) was focused on PP, meningitis, bacteraemia and AOM emergency visits from 2012-2018. These results cannot be compared to our primary care outcomes of interest. 

Comment 11: 10. Wouldn’t it have been logical to expect increasing cost to decrease over the study period for IPD, PP and ACP?

Author response: Thank you for your comment. According to the limited literature available regarding IPD, PP and ACP costs, no strong tendency has been seen in the most recent years. In our study no significant cost decrease was seen in the inpatient setting for IPD, PP and ACP. In contrast, from 2012 to 2018 in Italy, Amicizia et al (https://doi.org/10.1080/21645515.2022.2082205) observed an opposite trend in emergency department visits - an increase in PP, ACP ED visits hospitalization costs, and a decrease in AOM ED visits hospitalization costs.

Comment 12: 11. Finally, the authors obviously need to proof-read the manuscript. “Error! Reference source not found” on several places does not reflect meticulous proof-reading before submitting.

Author response: Thank you for your comment, we have reviewed and remove all cross-references along the manuscript.

---

## [Decision Letter · Decision Letter 1]

2 Mar 2023

Healthcare resource utilisation and cost of pneumococcal disease from 2003 to 2019 in children ≤17 years in England

PONE-D-22-26429R1

Dear Dr. Salini Mohanty,

We’re pleased to inform you that your manuscript has been judged scientifically suitable for publication and will be formally accepted for publication once it meets all outstanding technical requirements.

Kind regards,

Rahul Garg, MD

Academic Editor

PLOS ONE

Additional Editor Comments (optional):

Reviewers' comments:

Reviewer's Responses to Questions

**Comments to the Author**

1. If the authors have adequately addressed your comments raised in a previous round of review and you feel that this manuscript is now acceptable for publication, you may indicate that here to bypass the “Comments to the Author” section, enter your conflict of interest statement in the “Confidential to Editor” section, and submit your "Accept" recommendation.

Reviewer #2: All comments have been addressed

2. Is the manuscript technically sound, and do the data support the conclusions?

Reviewer #2: Yes

3. Has the statistical analysis been performed appropriately and rigorously? 

Reviewer #2: Yes

4. Have the authors made all data underlying the findings in their manuscript fully available?

Reviewer #2: Yes

5. Is the manuscript presented in an intelligible fashion and written in standard English?

Reviewer #2: Yes

6. Review Comments to the Author

Reviewer #2: The authors have responded to all comment in satisfactory way.

In my view, the paper can be published.

7. PLOS authors have the option to publish the peer review history of their article (what does this mean?). If published, this will include your full peer review and any attached files.

Reviewer #2: No

---

## [Editor Report · Acceptance letter]

8 Mar 2023

PONE-D-22-26429R1 

Healthcare resource utilisation and cost of pneumococcal disease from 2003 to 2019 in children ≤17 years in England 

Dear Dr. Mohanty:

I'm pleased to inform you that your manuscript has been deemed suitable for publication in PLOS ONE. Congratulations! Your manuscript is now with our production department. 

Kind regards, 

on behalf of

Dr. Rahul Garg 

Academic Editor

PLOS ONE